# CrohnDB: A Web Database for Expression Profiling of Protein-Coding and Long Non-Coding RNA Genes in Crohn Disease

**Rebecca Distefano** [1], **Mirolyuba Ilieva** [2], **Jens Hedelund Madsen** [2] **and Shizuka Uchida** [2,*]

[1] Department of Biology, University of Copenhagen, DK-2200 Copenhagen N, Denmark; fwh492@alumni.ku.dk
[2] Center for RNA Medicine, Department of Clinical Medicine, Aalborg University, DK-2450 Copenhagen SV, Denmark; mirolyubasi@dcm.aau.dk (M.I.); jenshm@dcm.aau.dk (J.H.M.)
* Correspondence: heart.lncrna@gmail.com or suc@dcm.aau.dk

**Abstract:** Crohn disease (CD) is a type of inflammatory bowel disease that causes inflammation in the digestive tract. Cases of CD are increasing worldwide, calling for more research to elucidate the pathogenesis of CD. For this purpose, the usage of the RNA-sequencing (RNA-seq) technique is increasingly appreciated, as it captures RNA expression patterns at a particular time point in a high-throughput manner. Although many RNA-seq datasets are generated from CD patients and compared to those of healthy donors, most of these datasets are analyzed only for protein-coding genes, leaving non-coding RNAs (ncRNAs) undiscovered. Long non-coding RNAs (lncRNAs) are any ncRNAs that are longer than 200 nucleotides. Interest in studying lncRNAs is increasing rapidly, as lncRNAs bind other macromolecules (DNA, RNA, and/or proteins) to finetune signaling pathways. To fill the gap in knowledge about lncRNAs in CD, we performed secondary analysis of published RNA-seq data of CD patients compared to healthy donors to identify lncRNA genes and their expression changes. To further facilitate lncRNA research in CD, we built a web database, CrohnDB, to provide a one-stop-shop for expression profiling of protein-coding and lncRNA genes in CD patients compared to healthy donors.

**Keywords:** Crohn disease; gene expression; inflammatory bowel disease; lncRNA; RNA-seq

## 1. Introduction

Crohn disease (CD) is a type of inflammatory bowel disease that causes chronic inflammation in the digestive tract [1–3]. The symptoms of CD include abdominal pain, cramping, fatigue, malnutrition, severe diarrhea, and weight loss. The number of CD patients is increasing worldwide, with 322 cases per 100,000 individuals in Germany and 319 cases per 100,000 individuals in Canada [4]. Although diet and stress have been suspected as potential causes of CD [5–7], the exact cause remains unknown [2]. Thus, intensive research has been underway to identify the causes of CD. Among the numerous approaches that have been taken so far, the unbiased, high-throughput techniques used to analyze DNA, RNA, and proteins are becoming increasingly popular [8–12]. As with any other multifactorial diseases (e.g., cardiovascular disease and diabetes), it is difficult to pinpoint the exact mechanisms underlying CD. This challenge is further complicated by the lack of systematic analysis of high-throughput data across different studies, followed by functional validation experiments.

The term junk DNA [13] has become outdated, as it is now clear that most of the human genome is transcribed as RNA [14]. However, only a small fraction of the transcribed RNAs correspond to the exons of protein-coding messenger RNAs (mRNAs) [15], with the majority of them being non-protein-coding and collectively known as non-coding RNAs (ncRNAs). In addition to classical examples of ncRNAs (e.g., ribosomal RNAs (rRNAs) and

transfer RNAs (tRNAs)), recent studies have highlighted the potential functional importance of long ncRNAs (lncRNAs) [16]. By definition, lncRNAs are any non-protein coding RNAs that are longer than 200 nucleotides (nt) [17]. Based on this broad definition, the current estimated number of lncRNAs is much higher than that of protein-coding genes [18]. Similar to the functions of proteins, the functions of lncRNAs are diverse, and while some have been discovered to play critical roles in various biological processes, the functionality of many remains unknown. The known functions of lncRNAs are mostly mediated through their binding partners, such as DNA, RNA, and proteins, to regulate epigenetic, transcriptional, post-transcriptional, and translational activities [19–21]. As most lncRNAs are expressed in a cell- and tissue-specific manner, as well as being time-dependent [22], the dysregulation of lncRNAs is observed in disease states [23,24] (e.g., CD [25,26]) compared to healthy donors, regardless of whether such lncRNAs are functionally important to the pathogenesis of a particular disease.

Here, we performed a secondary analysis of published RNA-sequencing (RNA-seq) data of patients suffering from CD compared to healthy donors to identify CD-related lncRNAs. To facilitate the further characterization of lncRNAs in CD, we built a web database, CrohnDB (https://rebeccadistefano.shinyapps.io/CrohnDB/, accessed on 22 May 2022), to offer a one-stop-shop for RNA expression profiles of both protein-coding and lncRNA genes in CD patients compared to healthy donors.

## 2. Materials and Methods

### 2.1. RNA-Seq Data Analysis and Visualization

As previously performed [27,28], RNA-seq data were downloaded from the Sequence Read Archive (SRA) database using the SRA Toolkit [29]. After expansion and conversion of SRA files, FASTQ files were preprocessed with fastp [30] (versions 0.21.0 and 0.22.0) using default settings to perform quality control, trim sequencing adapters, filter according to read quality, and carry out read pruning. The trimmed reads were mapped to the reference genome (GRCh38.107), using STAR [31] (versions 020201 and 2.7.9a).

Currently, there are different normalized values for RNA expressions, with the most common being counts per million (CPM), fragments per kilobase of exon per million mapped fragments (FPKM), reads per kilobase of transcript per million mapped reads (RPKM), and transcripts per kilobase million (TPM). For this study, CPM values were used to avoid issues related to alternative splicing and isoform length, as we recently demonstrated [27,28].

The R package, edgeR [32] (versions 3.30.3 and 3.32.1), was used to calculate counts per million (CPM) values and to derive differentially expressed genes. False discovery rate (FDR)-adjusted *p*-values were used for further analysis, unless stated otherwise in the text. The commands and programs used in this study can be found on the GitHub repository (https://github.com/heartlncrna/Analysis_of_CD_Studies; accessed on 23 March 2023).

The R-package, ggplot2 [33], was used to plot violin and volcano plots after removing genes with zero CPM values. To draw Venn diagrams, http://bioinformatics.psb.ugent.be/webtools/Venn/ (accessed on 6 September 2022) was used. The Database for Annotation, Visualization, and Integrated Discovery (DAVID) (version v2022q2) [34,35] was used to analyze for gene ontology (GO) terms and the Kyoto Encyclopedia of Genes and Genomes (KEGG) pathways.

### 2.2. CrohnDB Web Database

The CrohnDB web database was created using the R package Shiny [36]. The application consists of two main components: a user interface (UI) object and a server function. The former contains the R code that controls the appearance of the app, while the server function builds the application by loading the data, transforming the data, and connecting the inputs introduced in the UI object to specific outputs. These are then changed, based on users' inputs, through reactive programming, which updates the application instantly

if changes are made in the user interface. However, to reduce setup time, every input parameter has a default value.

CrohnDB has three main pages: (1) the Explore page; (2) the Download page; and (3) the Documentation page. The Explore page is the main page of the database. Here, the user can dynamically select the study to analyze, which will prompt the app to load the correct dataset into the database and store it into a data frame, which will be displayed through an interactive Result Table, rendered using the R package DT (https://github.com/rstudio/DT; accessed on 4 April 2023). This action will, in turn, update the "Select Comparison" input. Differentially expressed genes (DEGs) will then be identified based on the users' "logFC" (fold-change value in a logarithm of base two) and "FDR" inputs, distinguishing between lncRNA and protein-coding genes (Figure 1). DEGs can be further explored on the right hand-side of the Explore page, which is divided into five tabs: (1) Volcano Plot; (2) Heatmap; (3) GO Analysis; (4) Pathway Analysis; and 5) Comparisons Intersection.

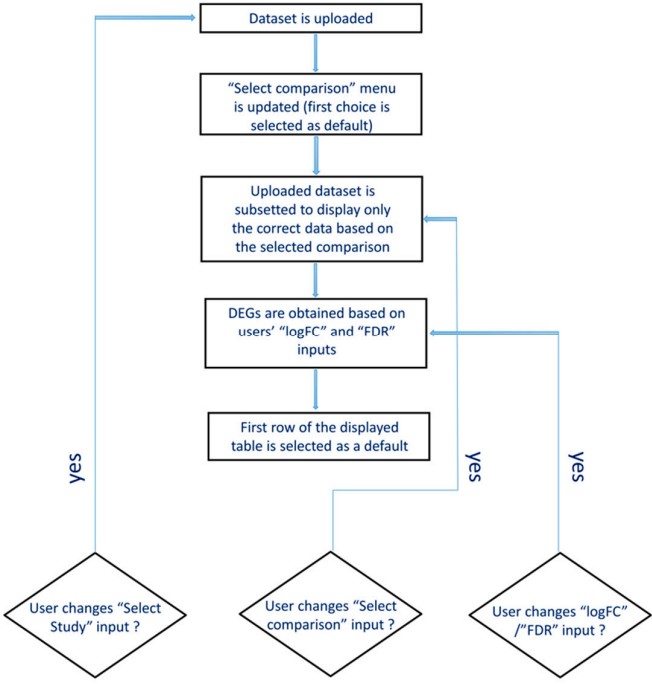

**Figure 1.** Workflow of initial data processing for the Explore page setup in CrohnDB.

The volcano plot is displayed using the R package ggplot2 [33]. Here, an observeEvent object and the reactive programming functionality of the Shiny library allow the user to select a row from the Result Table, which will, in turn, update the volcano plot to display the gene in the selected row as a dot that is larger than the rest of the dots to highlight the target gene in the volcano plot. Furthermore, a boxplot rendered using the ggplot2 andplotly [33,37] will display the distribution of the selected gene based on CPM values, while distinguishing between the two conditions being compared. In addition, a small summary table with the numbers of dysregulated lncRNA and protein-coding genes is generated using the renderTable function. The Heatmap tab detects the input in the "Select Comparison" drop-down menu and loads the corresponding file with the CPM values of the chosen dataset, which is then stored as a data frame. DEGs are then identified, based on user selected "logFC" and "FDR" threshold values. The resulting data frame is further subsetted, based on the users' choice of gene biotype, and orderedbased on threshold values (FDR values and absolute logFC). After this process, the top 30 genes are selected and displayed in a heatmap rendered using the R packages ggplot2 and plotly [33,37]. The GO analysis is performed through the R package gprofiler2 (using only the categories "GO:BP", "GO:MF" and "GO:CC") [38] after the data frame has been subsetted based on the user selected expression pattern, with the results visualized both

as a dotplot created using gprofiler2 and plotly [37–39] and as a table, rendered using the R package DT. Pathway analysis is performed using the function enrichKEGG from the clusterProfiler R package [39] after the data frame has been subsetted, based on the user selected expression pattern, and visualized as a dotplot, rendered using the R package enrichplot [40]. Finally, in the Comparisons Intersection tab, the user can choose between displaying a Venn diagram rendered using the R package euler [41] or an upset plot rendered using the UpSetR R package [42]. Here, based on the user's choice of diagram and expression pattern, a reactive list of up-regulated or down-regulated genes is temporarily created and used in the plot functions.

The Download page allows the user to select the desired dataset to be downloaded from the drop-down menu. This will prompt the app to upload the correct file, which will be saved as a data frame and displayed in a table though the R package DT. The desired dataset can then be downloaded in either comma-separated values (csv) or tab-separated values (tsv) format.

All the data loaded into the database are dynamically removed after the user has finished interacting with such data, to reduce the random-accessed memory (RAM) needed to run the app on the server.

The CrohnDB database was designed and developed by adhering to the principles of the findability, accessibility, interoperability, and reusability (FAIR) of scientific data [42]. All the datasets used to build the CrohnDB database are persistently referred to using the Gene Expression Omnibus (GEO) ID. All information is described in the Documentation page, where the link to the GEO homepage can also be found, making the data findable. Furthermore, CrohnDB is freely available online, without a password, from https://rebeccadistefano.shinyapps.io/CrohnDB/ (accessed on 8 April 2023), where the data can be easily accessed and downloaded. To further facilitate data reuse and collaboration, all the data files used for the database are also available in the GitHub repository https://github.com/Reb08/CrohnDB (accessed on 4 April 2023). To ensure interoperability, the data have been annotated using common ontology terms, analyzed with standard thresholds, and can be downloaded in standard formats (.tsv and .csv). Finally, the data are reusable, as the data are well described in the Documentation page. In addition, comprehensive documentation on data processing can be found in the GitHub repository https://github.com/heartlncrna/Analysis_of_CD_Studies (accessed on 8 May 2023), while data used to generate CrohnDB are available in the GitHub repository: https://github.com/Reb08/CrohnDB (accessed on 4 April 2023).

## 3. Results

### 3.1. Hundreds of Genes Are Differentially Expressed in CD Patients Compared to Healthy Donors

Although CD can affect any levels of the intestinal tract, from the mouth to the anus, over 60% of CD patients have colonic involvement [43]. Thus, to examine the expression profiles of CD patients compared to healthy donors, RNA-seq data generated from colon tissues were analyzed. This dataset [GEO accession number, GSE164871] contains RNA-seq data from four CD patients and four healthy donors. As expected [44], overall, lncRNA genes were lowly expressed compared to protein-coding genes (Figure 2A). CD patients were then compared to healthy donors using threshold values of two-fold and a false discovery rate (FDR) corrected p-value of less than 0.05, which identified 792 up-regulated and 188 down-regulated genes with 51 up-regulated and 39 down-regulated lncRNA genes (Figure 2B,C). When these differentially expressed genes were further analyzed, Gene Ontology (GO) terms related to immune and inflammatory responses, as well as phagocytosis, were enriched.

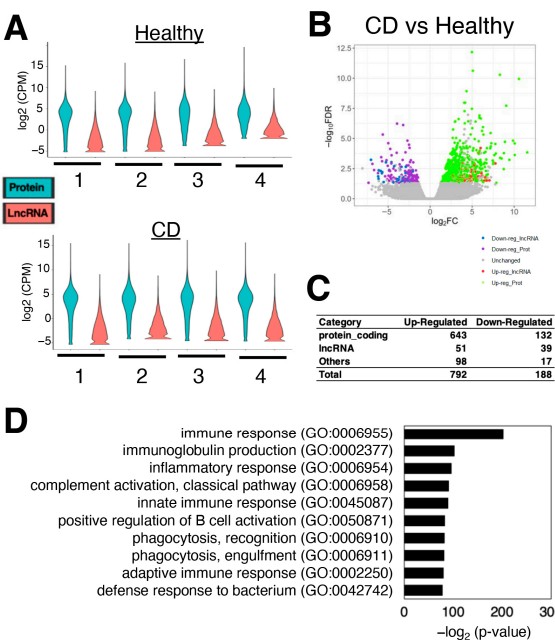

**Figure 2.** RNA-seq data analysis of colon tissues from four CD patients and four healthy donors. (**A**) The average CPM (counts per million) values for each sample were used to draw a violin plot for protein-coding and lncRNA genes based on the annotation (biotype) provided by the Ensembl database (GRCh38.107). (**B**) Volcano plots. The threshold values of two-fold and FDR < 0.05 were applied. To highlight up-regulated or down-regulated protein-coding (Prot) and lncRNA genes, different colors were used, as specified in the figure. Not differentially expressed genes (unchanged) are shown in gray. (**C**) Numbers of differentially expressed protein-coding, lncRNA, and other genes. Other genes include microRNAs (miRNAs), pseudogenes, and ribosomal RNAs (rRNAs), based on the annotation provided by the Ensembl database. (**D**) Top 10 enriched GO terms for 980 differentially expressed genes.

As with any other disease, the severity of CD varies significantly among patients. To further dissect lncRNA expression patterns in CD patients, the RNA-seq dataset profiling CD patients based on clinical phenotypes was reanalyzed (GEO accession number, GSE66207). Peck et al. conducted small RNA-seq and RNA-seq experiments using colon tissues from five nonstricturing and nonpenetrating (B1), six stricturing (B2), and eight penetrating/fistulizing (B3) CD patients compared to 13 control subjects without CD [45]. The authors stated: "Stricturing disease was defined as the occurrence of constant luminal narrowing demonstrated by radiologic, endoscopic, or surgical examination combined with prestenotic dilatation and/or obstructive signs or symptoms. Penetrating disease was defined as the presence of perianal, enteroenteric, or enterovesicular fistulae, intra-abdominal abscesses or intestinal perforation." [45]. The main findings of that study included the following: (1) a set of miRNAs (e.g., *miR-31-5p*, *miR-215*, *miR-223-3p*, *miR-196b-5p*, and *miR-203*) can be used to stratify CD patients based on disease behavior independent of the effect of inflammation; (2) the expression of *miR-215* indicates the likelihood of progression to penetrating/fistulizing CD; and (3) *miR-31* and *miR-203* were identified as candidate master regulators of signaling pathways disrupted in CD pathogenesis [45]. As the main focus of the authors was on miRNAs, the generated RNA-seq data were analyzed only for protein-coding genes, but not for lncRNAs.

When the threshold values of two-fold and FDR < 0.05 were applied, there were more differentially expressed genes identified in B2 and B3 than in B1 compared to the control subjects, as was reported in the original study (Figure 3A,B). To further characterize these differentially expressed genes, overlapping genes among the three comparisons were searched (Figure 3C). Although there were no overlapping down-regulated genes, there were four up-regulated protein-coding genes commonly shared among three comparisons: C-X-C motif

chemokine ligand 5 (*CXCL5*), hydroxy-delta-5-steroid dehydrogenase, 3 beta- and steroid delta-isomerase 2 (*HSD3B2*), nitric oxide synthase 2 (*NOS2*), and serum amyloid A2 (*SAA2*). However, no lncRNA gene was shared in up-regulated or down-regulated genes.

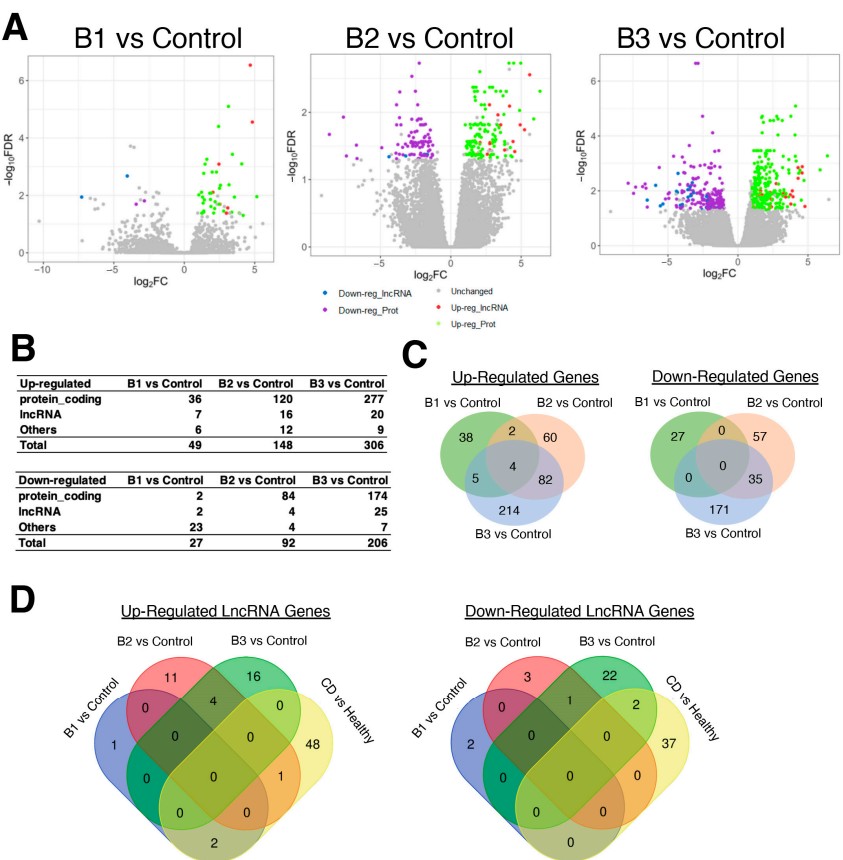

**Figure 3.** RNA-seq data analysis of colon tissues from five B1, six B2, and eight B3 CD patients compared to 13 control subjects without CD. (**A**) Volcano plots. The threshold values of two-fold and FDR < 0.05 were applied. (**B**) Numbers of differentially expressed protein-coding, lncRNA, and other genes. Other genes include miRNAs, pseudogenes, and rRNAs, based on the categories provided by the Ensembl database. (**C**) Venn diagrams of up-regulated and down-regulated genes for three comparisons. (**D**) Venn diagrams of up-regulated and down-regulated lncRNAs genes comparing two datasets (GSE164871 and GSE66207) for four comparisons.

Because we were interested in the dysregulation of lncRNAs in CD patients compared to healthy donors, differentially expressed lncRNA genes were compared between two datasets, consisting of four comparisons (Figure 3D). As shown in Figure 3C, no overlapping lncRNA gene was found among all four conditions. Instead, only seven up-regulated and three down-regulated lncRNA genes were shared between two comparisons (Supplementary Table S1). Among these shared differentially expressed lncRNA genes, only one lncRNA gene has been reported to date. The lncRNA gene is *LINC01819* (long intergenic non-protein coding RNA 1819), which is up-regulated in B1 compared to the controls and CD and healthy donors. This lncRNA gene is located on Chromosome 2: 43,027,823-43,040,662 and has 14 transcripts (isoforms). Recently, Zhang et al. reported *LINC01819* as one of six prognostic metastasis-associated lncRNAs in patients with lung adenocarcinoma [46]. However, *LINC01819* has not been studied in CD. Thus, further research is needed to uncover the roles of these shared differentially expressed lncRNAs in CD patients.

### 3.2. Severeal Differentially Expressed lncRNA Genes Are Shared in Fibroblasts Isolated from Different Etiologies of CD Patients Compared to Healthy Control Subjects

As CD is a type of inflammatory bowel disease, the association between inflammatory cells and fibrosis is of great interest, as fibrotic tissues cause a multitude of complications, not only in CD but also in other chronic inflammatory diseases, including cardiac, liver, and pulmonary fibrosis [47–49]. Yim et al. performed an RNA-seq experiment to profile fibroblasts isolated from the mucosa of terminal ileal tissue of CD patients [50]. The dataset (GEO accession number, GSE99816) contained RNA-seq data from six control subjects and 15 CD patients consisting of five macroscopically normal (non-inflamed; NINF), four inflamed (INF), and five stenotic (STEN) CD patients' samples. When the threshold values of two-fold and $p$-value < 0.05 were applied, several hundred genes were differentially expressed in all etiologies of CD patients compared to fibroblasts isolated from control subjects (Figure 4A,B).

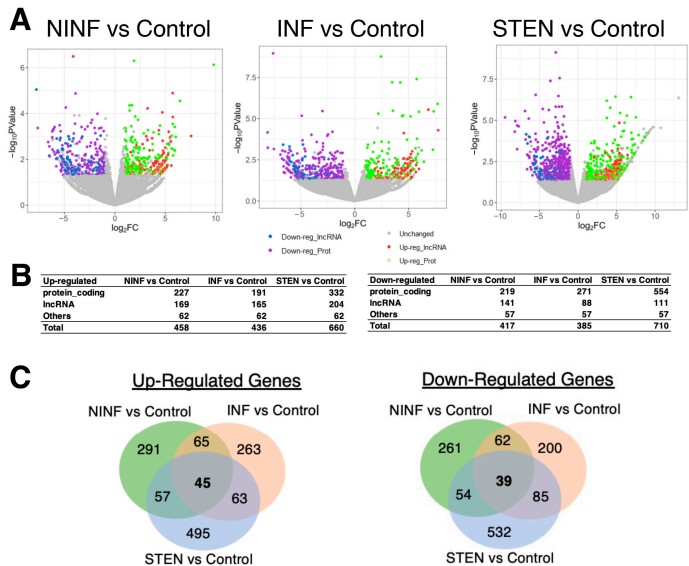

**Figure 4.** RNA-seq data analysis of five macroscopically normal (non-inflamed; NINF), four inflamed (INF), and five stenotic (STEN) CD patients' samples compared to six control subjects. (**A**) Volcano plots. The threshold values of two-fold and $p$-value < 0.05 were applied. (**B**) Numbers of differentially expressed protein-coding, lncRNA, and other genes. Other genes included miRNAs, pseudogenes, and rRNAs, based on the categories provided by the Ensembl database. (**C**) Venn diagrams of up-regulated and down-regulated genes showing all three comparisons.

When the overlapping genes among the etiologies were searched, there were 45 up- and 39 down-regulated genes shared among all three etiologies compared to the control, including 14 up-regulated and 10 down-regulated lncRNA genes (Figure 4C and Supplementary Table S2). Among these 24 shared differentially expressed lncRNAs, only one up- and three down-regulated lncRNAs were described in publications. One up-regulated lncRNA is *LINC01152* (long intergenic non-protein coding RNA 1152), which has been reported to promote cell proliferation and survival in hepatocellular carcinoma [51] and functions as an miRNA sponge in glioblastoma multiforme cells to sequester *miR-466* [52]. Three down-regulated lncRNA genes are *EPB41L4A-DT* (EPB41L4A divergent transcript), *HOXC-AS2* (HOXC cluster antisense RNA 2), and *LINC02323* (long intergenic non-protein coding RNA 2323). While *EPB41L4A-DT* is indicated as a potential biomarker of clear cell renal cell carcinoma based on its expression patterns [53,54], the other two lncRNAs are reported as miRNA sponges in different cancer types [55–58]. However, most other shared lncRNA genes have not been studied to date. Furthermore, all shared lncRNA genes have not been reported in the context of CD. This calls for further investigation.

### 3.3. The Web Database, CrohnDB, for Screening of Protein-Coding and lncRNA Genes

As only handful of lncRNAs are functionally studied in CD (e.g., *DQ786243* [59], *IRF1-AS1* [60], *MALAT1* [61], and *LINC01272* [62]), we built a web database, CrohnDB, to facilitate further research into protein-coding and lncRNA genes in CD (Figure 5A). From the Explore tab, the three studies analyzed above (GSE66207, GSE99816, and GSE164871) can be explored (Figure 5B). Users can flexibly set the threshold values for fold change in a logarithm of base two scale and FDR or p-values (in the case of GSE99816) to identify differentially expressed genes (both protein-coding and lncRNA genes), whose numbers are dynamically generated in the table in this window. The top differentially expressed genes can be further examined for their expressions via a heat map (Figure 5C). In addition, these differentially expressed genes can be analyzed for enriched GO terms (Figure 5D) and KEGG pathways (Figure 5E), which can be selected from this window, to obtain a global view of differentially expressed gene functional categories and signaling pathways. Within each study, there are several comparisons of conditions available, which can be visualized via Comparisons Intersection (Figure 5F). In addition, all the available data can be downloaded from the Download window so that further processing of the analyzed data is possible.

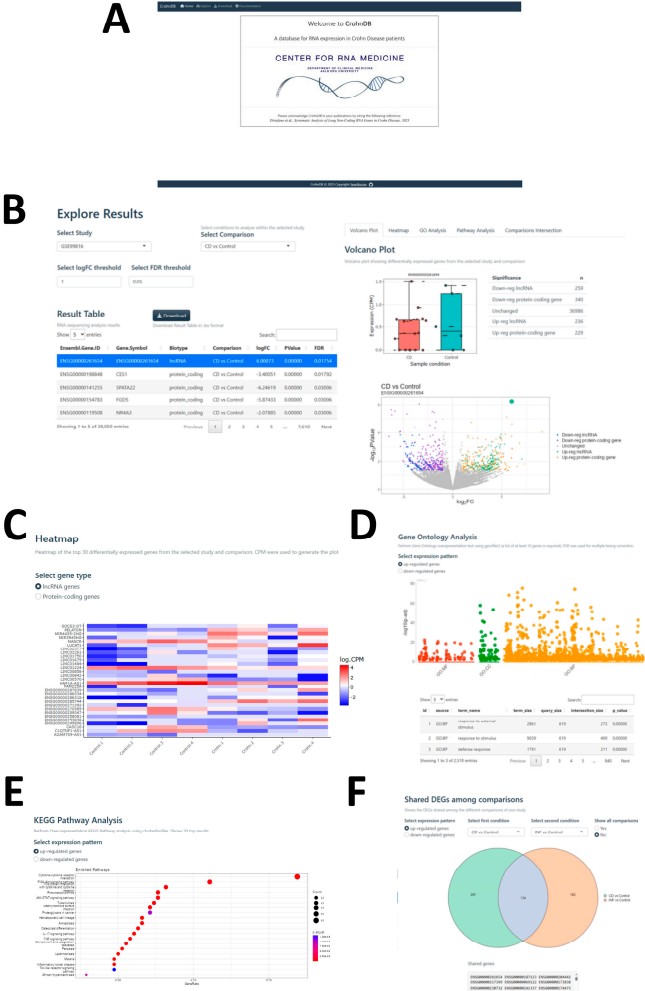

**Figure 5.** The CrohnDB web database. (**A**) The home page of CrohnDB. (**B**) Explore Results tab and window. Each differentially expressed gene can be inspected from the Result Table and the heat map. (**C**–**E**) The differentially expressed genes can be visualized via (**C**) the heat map and further analyzed for the enriched (**D**) GO terms and (**E**) pathways. (**F**) As there are several conditions within each study, differentially expressed genes among the different conditions to be compared can be visualized via the Comparisons Intersection tab.

## 4. Discussion

Here, we determined that many differentially expressed genes are identified in CD patients compared to healthy donors, including lncRNA genes. However, most CD-related lncRNA genes are not functionally or mechanistically studied. To facilitate further research in CD, especially for lncRNA genes, we built the web database CrohnDB to explore the expression profiles of protein-coding and lncRNA genes in published studies. With CrohnDB, users can easily screen for both protein-coding and lncRNA genes that are differentially expressed in CD patients compared to healthy donors to further design functional and mechanistic studies for differentially expressed genes, especially lncRNA genes. While CrohnDB allows users to interactively choose different threshold values, biotypes, and expression patterns to focus the analysis, some compromises were made to find the right balance between flexibility and ease of use. To reduce the complexity of user experience, no choice of biotype was added in the Volcano plot tab, while the FDR threshold value to determine significantly enriched terms in GO and KEGG analyses was set to 0.05, without the possibility of changing it. Furthermore, to reduce loading time and RAM usage, the Result table displayed only protein-coding and lncRNA genes, without other ncRNAs (e.g., rRNAs, tRNAs). In the future, we aim to regularly update software and packages to ensure that the app remains secure and up-to-date, as well as updating the database when new RNA-seq data related to CD become available. In addition, to ensure a user-friendly experience, we will regularly monitor and promptly address user feedback, which can be provided at https://github.com/Reb08/CrohnDB/issues (accessed on 8 May 2023). To this end, regular tests will also be performed to ensure the app remains functional, while expanding functionality and documenting changes and updates, to ensure replicability.

As with any other studies, our current study has limitations. First, all RNA-seq data analyzed here were generated from RNAs with poly A tails. Given that more than half of lncRNAs do not have poly A tails [63], we underestimated the number of CD-related lncR-NAs. Second, *p*-values used to derive differentially expressed genes from the RNA-seq data of isolated fibroblasts comparing between CD patients and control subjects were not corrected for multiple tests (i.e., FDR-correction). This was because fibroblasts are well known for their heterogeneity [64]. Thus, sample variabilities were very high, resulting in limited numbers of differentially expressed genes, with three, 16, and 38 differentially expressed genes in NINF, INF, and STEN samples, respectively, compared to the control samples.

**Supplementary Materials:** The following supporting information can be downloaded at: https://www.mdpi.com/article/10.3390/computation11060105/s1, Supplementary Table S1. List of shared differentially expressed lncRNA genes comparing two datasets (GSE164871 and GSE66207) for four comparisons; Supplementary Table S2. List of shared differentially expressed genes among three comparisons.

**Author Contributions:** Conceptualization, S.U.; methodology, R.D. and S.U.; database, R.D.; validations, M.I. and J.H.M.; resources, S.U.; writing—original draft preparation, R.D. and S.U. All authors have read and agreed to the published version of the manuscript.

**Funding:** This work was supported by grants from the Department of Clinical Medicine, Aalborg University (to S.U.).

**Data Availability Statement:** The Supplementary Materials can be found on the GitHub repository: https://github.com/heartlncrna/Analysis_of_CD_Studies (accessed on 23 March 2023). All codes used to generate CrohnDB are available on the GitHub repository: https://github.com/Reb08/CrohnDB (accessed on 4 April 2023).

**Conflicts of Interest:** The authors declare no conflict of interest.

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
