# Peer review of "CrohnDB: A Web Database for Expression Profiling of Protein-Coding and Long Non-Coding RNA Genes in Crohn Disease"

_computation, doi:10.3390/computation11060105_

Round 1

Reviewer 2 Report

The article entitled “CrohnDB: a Web Database for Expression Profiling of Protein coding and Long non-Coding RNA Genes in Crohn Disease” by Distefano et al, document a database that utilizes multiple public domain dataset to illustrate the significance of protein-coding genes and lncRNAs in Crohn disease. Overall, the database is simplified for the users to scan the resulting outcomes and relate them to individual studies. It could be an important resource for the researchers to explore their genes of interest in varying cohorts.

I have a few concerns about the database as below:

This database includes multiple studies related to Crohn's disease and its complications. However, no attempts were made to comprehend the status of differentially expressed genes or lncRNAs across them. For instance – The authors could have expanded the elements in the “Comparisons Intersection” tab to illustrate the common/unique features as a Venn diagram and further be viewed as a table for these dissected genes.

Similarly, the database should comprehensively demonstrate and recommend the researchers, top X genes or lncRNAs that are crucial to Crohn's disease based on condensing all the datasets ( or a subset) utilized in this study.

In general, the user interface can be reorganized and aligned properly with all the elements. Also, the font size of tables can be reduced accordingly.

For each cohort selected, the expression profile of highlighted gene (in the volcano plot) should be accompanied by a Violin plot. This will help to quickly review the median occurrence and statistical significance of genes in the given cohort (along with the displayed table).

Authors are requested to enable the “hover” option or label the feature wherever possible. For instance: a) In the heatmap, it is not clear what we are the feature we are looking at on the Y-axis. If require expand the heatmap to make the y-axis readable. b) For GO analysis, please enable the hover option to display the processes for each bubble.
